# A Novel *AKT1*, *ERBB2*, *ESR1*, *KRAS*, *PIK3CA*, and *TP53* NGS Assay: A Non-Invasive Tool to Monitor Resistance Mechanisms to Hormonal Therapy and CDK4/6 Inhibitors

**DOI:** 10.3390/biomedicines12102183

**Published:** 2024-09-26

**Authors:** Alessandra Virga, Caterina Gianni, Michela Palleschi, Davide Angeli, Filippo Merloni, Roberta Maltoni, Paola Ulivi, Giovanni Martinelli, Ugo De Giorgi, Sara Bravaccini

**Affiliations:** 1Biosciences Laboratory, IRCCS Istituto Romagnolo per lo Studio dei Tumori (IRST) “Dino Amadori”, 47014 Meldola, Italy; paola.ulivi@irst.emr.it (P.U.); sara.bravaccini@irst.emr.it (S.B.); 2Department of Medical Oncology, IRCCS Istituto Romagnolo per lo Studio dei Tumori (IRST) “Dino Amadori”, 47014 Meldola, Italy; caterina.gianni@irst.emr.it (C.G.); michela.palleschi@irst.emr.it (M.P.); filippo.merloni@irst.emr.it (F.M.); roberta.maltoni@irst.emr.it (R.M.); ugo.degiorgi@irst.emr.it (U.D.G.); 3Unit of Biostatistics and Clinical Trials, IRCCS Istituto Romagnolo per lo Studio dei Tumori (IRST) “Dino Amadori”, 47014 Meldola, Italy; davide.angeli@irst.emr.it; 4Department of Hematology and Sciences Oncology, Institute of Haematology “L. and A. Seràgnoli”, S. Orsola University Hospital, 40138 Bologna, Italy; giovanni.martinelli2@unibo.it

**Keywords:** hormone receptor-positive metastatic breast cancer patients, CDK4/6 inhibitors resistance, novel multi-gene target panel NGS assay

## Abstract

**Background**: Patients with hormone receptor-positive (HR+)/HER2- metastatic breast cancer (mBC) generally receive hormonal therapy (HT) combined with CDK4/6 inhibitors (CDK4/6i). Despite this treatment, resistance mechanisms to CDK4/6i emerge and the majority of these patients experience disease progression (PD). This highlight the necessity to uncover the resistance mechanism to CDK4/6i through the identification of specific biomarkers. The primary objective is to assess the accuracy and feasibility of a novel multi-gene target panel NGS assay on circulating tumor DNA (ctDNA) to detect molecular alterations of *AKT1*, *ERBB2*, *ESR1*, *KRAS*, *PIK3CA*, and *TP53* genes in women with BC undergoing HT plus CDK4/6i treatment. Secondarily, the study aims to explore the relationship between genomic profiling and clinical outcomes. **Materials and Methods**: Plasma samples were collected from 16 patients diagnosed with advanced/locally advanced HR+/HER2- BC at 2 time points: T0 (baseline) and at T1 (3 months after CDK4/6i treatment). Starting from 2 mL of plasma, ctDNA was isolated and libraries were set up using the Plasma-SeqSensei (PQS)^®^ Breast Cancer IVD Kit, sequenced on Nextseq 550 and analyzed using the Plasma-SeqSensei™ IVD Software^®^. **Results**: Among the five patients who presented PD, three had *PIK3CA* mutations and, of these, two showed a higher mutant allele frequency (MAF) at T1. In three patients with stable disease and in eight patients with partial response, the MAF of the detected alterations decreased dramatically or disappeared during CDK4/6i treatment. **Conclusions**: Based on our findings, the liquid biopsy analysis using the PQS panel seems to be both feasible and accurate, demonstrating a strong sensitivity in detecting mutations. This exploratory analysis of the clinical outcome associated to the mutational status of patients highlights the potential of molecular analysis on liquid biopsy for disease monitoring, although further validation with a larger patient cohort is necessary to confirm these preliminary observations.

## 1. Introduction

Breast cancer (BC) is the most commonly diagnosed cancer and is a heterogeneous disease, classified into many subgroups according to the presence of hormone receptors, Ki-67, and HER2 expression [1]. About 70% of BCs are estrogen receptor (ER)- and progesterone receptor (PgR)-positive. Endocrine therapy is the gold standard of treatment; however, around 30–50% of patients with BC will relapse [2]. Around 15–20% of ER+ BC patients are intrinsically resistant to treatment, and an additional 30–40% become resistant during treatment. Patients exhibiting resistance show elevated mutation rates and the presence of resistant subclones, suggesting an increased risk of mortality or disease recurrence [3,4,5,6]. Recently, the introduction of CDK4/6 inhibitors (CDK4/6i) has significantly improved the outcome for patients with hormone receptor-positive (HR+) BC in advanced stages, by targeting the cell cycle machinery [4,5]. Approved drugs such as palbociclib, ribociclib, and abemaciclib are now used together with aromatase inhibitors or fulvestrant for the treatment of HR+ metastatic BC patients. Despite these advancements, a substantial number of patients develop disease progression (PD), highlighting the urgent need to explore the underlying mechanisms of resistance to CDK4/6i. The primary objective of this study is to assess the feasibility and accuracy of a new multi-gene next-generation sequencing (NGS) panel designed to identify genomic alterations in *AKT1*, *ERBB2*, *ESR1*, *KRAS*, *PIK3CA*, and *TP53* genes in circulating tumor DNA (ctDNA) extracted from the plasma of women receiving hormonal therapy (HT) and CDK4/6i. Furthermore, we aim to explore the relationship between genomic profiling and patient clinical outcomes.

## 2. Materials and Methods

Plasma samples were retrospectively collected from 16 patients with advanced/locally advanced HR+/HER2- BC, enrolled at IRCCS Istituto Romagnolo per lo Studio dei Tumori (IRST) “Dino Amadori” (Meldola, Italy). The cohort of patients was selected based on CDK4/6i treatment and in relation to response after 3 months of treatment. The samples were collected at two time points: at baseline (T0) and after three months of CDK4/6i treatment (T1). CtDNA was isolated from 2 mL of plasma using QIAamp Circulating Nucleic Acid Kit (Qiagen, Hilden, Germany) according to the manufacturer’s instructions. Plasma-Seqsensei (PQS)^®^ Breast Cancer IVD Kit (Sysmex Inostics, GmbH, Hamburg, Germany) was used for library preparation, which covers 3623 different COSMIC mutations within selected hotspots (3070 in *TP53*), and sequenced using Illumina NEXTSEQ550 platform (Illumina, San Diego, CA, USA). The NGS data were analyzed using Plasma-SeqSensei™ IVD Software^®^ (v1.2.2) (Appendix A). Sequences were aligned to the human reference genome hg19 (GRCh37), and an average read of 21765306 assigned to UID families with sufficient size was reached. The software classified the alterations according to somatic or germline origin. Mutations were listed by the software as potential germline mutations when they were present at an MAF >40% to ≤60% (heterozygous) or ≥90% (homozygous). Box plots and KM plots were generated using SRplot [7].

## 3. Results

Starting from a very low input of DNA (average 15.7 ng), 96.9% of the sequenced samples passed the four essential parameters (positive and negative controls, sequencing depth, and DNA quantification) quality control (QC) of the Plasma-SeqSensei™ IVD Software^®^. The test achieved a sensitivity of 95.2% and an optimal percentage in terms of specificity (Appendix A). A robust quantification of tumor-specific sequences over a broad dynamic range was achieved using the PQS technology with an internal quantifier, which allowed a detection limit of up to six mutant molecules (MM). Of 16 patients, 5 presented PD, 3 presented stable disease (SD), and 8 presented partial response (PR) after 3 months of CDK4/6i, as shown in Table 1.

Interestingly, of the five patients with PD, three had mutations in *PIK3CA* and one in *ESR1* genes in the T0 sample, as described in Figure 1A. Of these three *PIK3CA*-mutated patients with PD, two exhibited an increased mutant allele frequency (MAF) or MM in their T1 samples. One of these patients showed a similar trend in terms of MAF improvement for both *PIK3CA* and *ERBB2* mutations, while the other patient developed 2 novel *TP53* mutations during treatment (Figure 1A,B and Appendix A).

On the contrary, patients with SD and PR showed a significant decrease in the MAF of T1; in particular, the *AKT1* and the *ESR1* mutations disappeared during treatment and the same trend was observed for *TP53* and *PI3KCA* (Figure 1B, Appendix A). The genomic equivalent calculated by the software, showed a significant decrease in patients with SD or PR, compared to the patients with PD. Interestingly, one patient with PD was wild-type (WT) for the analyzed genes, but the GE increased from T0 to T1 (Figure 1B). The patient in PR with *PI3KCA* alteration showed three novel *ESR1* mutations in the plasma sample collected during treatment. Finally, patients with alterations in *PIK3CA* were significantly associated with worse PFS, and the same trend was observed in the OS of the BC population treated with HT in online resources (Appendix A). In addition, we performed a PFS analysis of the case series classified into different histotypes; lobular subtypes showed the best outcome (Figure 1C).

## 4. Discussion and Conclusions

The recent exploratory analyses of the PALOMA-3 trial indicate that OS improvement was maintained (>6 years of follow up) in HR+/HER2- advanced BC patients treated with palbociclib and fulvestrant, supporting this combination as a standard of care for these patients [8]. While the introduction of CDK4/6i (palbociclib, ribociclib, and abemaciclib) has led to improved progression-free survival in HR+/HER2- BC patients, research has shown that the presence of mutations involved in resistance pathways can result in reduced sensitivity to the related drugs [4,5,6]. Then, there is the need for the identification of molecular biomarkers that may be able to select responsive patients. The PADA-1 trial was the first trial that demonstrated the clinical utility of monitoring *ESR1* mutations in liquid biopsy, with a doubled PFS in *ESR1*-mutated patients who switched from AI-palbociclib to fulvestrant–palbociclib [9]. For the first time in Italy, we analyzed the accuracy of this novel NGS target panel assay (*AKT1*, *ERBB2*, *ESR1*, *KRAS*, *PIK3CA*, and *TP53*) on the plasma ctDNA of mBC patients treated with HT plus CDK4/6i. In the present study, we identified four alterations in *PIK3CA*, making it the most altered gene. Interestingly, as expected, three out of four of the *PIK3CA* mutations were found in patients with PD. We analyzed the survival trend of our entire cohort, considering alterations in *PIK3CA* as well as a larger online dataset of patients treated with HT. In both cases, the trend of response in patients with *PIK3CA* mutations was worse compared to those with WT *PIK3CA.* Consistent with the literature data, approximately 40% of HR+ BC cases exhibit activating mutations in the *PIK3CA* subunit of the PI3K gene, such as p.E545K, p.E542K, and p.H1047R [10,11]. Additionally, for the PD subgroup of patients, we observed a trend of improvement in the MAF value from T0 to T1, contrarily to patients with SD and PR. This trend in patients with PD involved both *PIK3CA* and *ERBB2* genes, supporting the association between treatment response and the increase in resistance alterations detected in liquid biopsies during patient monitoring [12,13]. For the patient with double mutations in *PIK3CA* and *ERBB2* genes, the increase in MAF during HT plus CDK4/6i may have significant and immediate clinical implications due to the novel targeted therapies available for HER2+ BC, such as trastuzumab deruxtecan [14,15].

According to MAF value changes, the GE values (3.3 pg of human DNA) also followed the same trend both in the PD and in the SD/PR subgroups of patients, characterized by a consistent increase and decrease in GE values, respectively. Notably, the only patient with PD and an absence of gene alterations was characterized by an increase in GE amount from T0 to T1. The literature data show that ctDNA fluctuations are associated with treatment response [16,17]. Despite not analyzing the ctDNA fraction in relation to patient response, our pilot study suggests that changes in GE values may also indicate treatment response during monitoring. Despite the significant decrease in MAF and GE values for SD/PR subgroups, 67% of patients with SD and 50% of patients with PR showed alterations in the 6 genes analyzed. *KRAS* and *AKT1* mutations were present only in these subgroups, with MAF values of 1.35% and 8.91%, respectively. Both *KRAS* and *AKT1* alterations, as a part of the PI3K/AKT/mTOR signaling transduction pathway, are associated with resistance mechanisms that arise during HT plus CDK4/6i treatments [18,19]. Nevertheless, these alterations disappeared during treatment, supporting the hypothesis of a good response to treatments, as reported in the literature for *KRAS* alterations [20]. We are aware that our panel permits the evaluation of only six genes; however, they are among the most frequently altered genes in patients who developed resistance mechanisms to HT plus CDK4/6i and for whom therapeutic drugs are available. Larger genomic analyses could help in understanding the interactions among the possible genes involved. Finally, consistent with literature data, our preliminary findings showed a better survival in patients with lobular BC than ductal BC. Despite this result, the assessment of molecular evaluation during treatments is necessary for ductal and lobular BC patients, considering the problem of the late recurrence also for the latter [21,22,23].

Considering the tumor heterogeneity and the small size of our population, testing for mutations in 6 genes may not be sufficient to have a full knowledge of all mechanisms of resistance given that other genes may be involved [24]. Despite these limitations, the liquid biopsy analysis using PQS was feasible and accurate. The assay passed several QC checks and permitted the detection of gene alterations up to 0.06% of the MAF value. Our exploratory secondary aim was to investigate the association between genomic alterations and clinical outcomes. Our findings suggest a trend of association between PD and SD/PR subgroups and the presence or absence of mutations. Consistent with literature data, the presence of *PIK3CA* alterations was associated with worse treatment response. Decreased and increased MAF values showed a promising role for disease monitoring. Our preliminary findings suggest that molecular analysis in liquid biopsy by using the novel NGS assay was feasible and accurate, and can be useful for a real-time evaluation of the disease. An enlarged case series of patients is needed to validate our preliminary results.

## Figures and Tables

**Figure 1 biomedicines-12-02183-f001:**
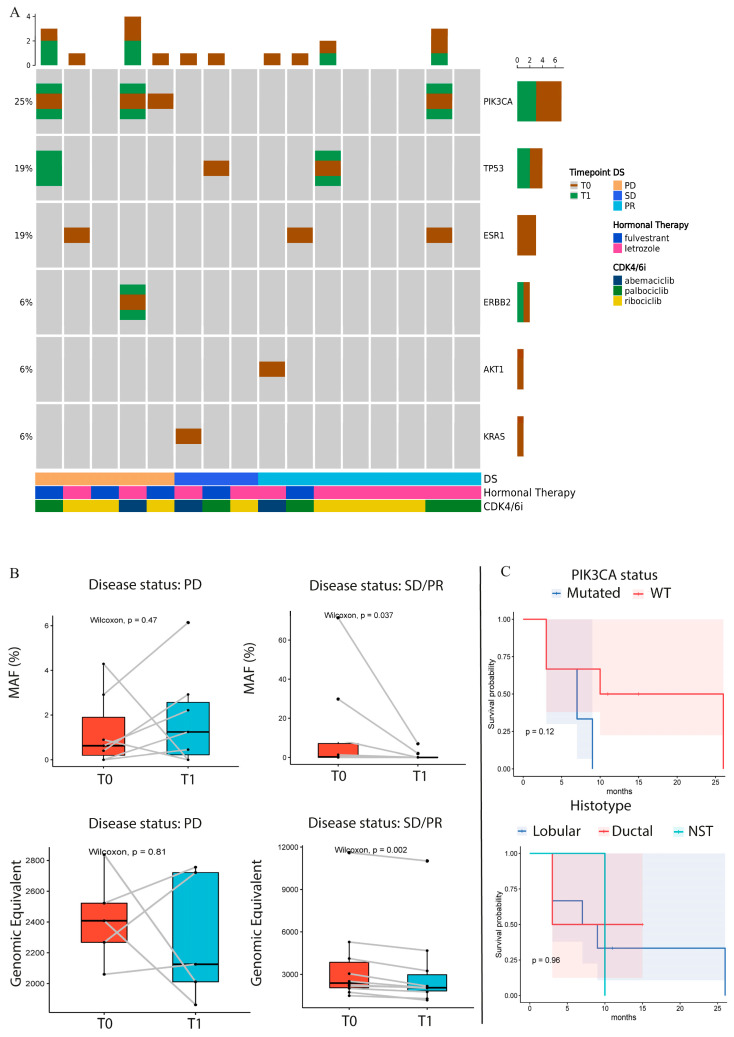
(**A**) Oncoprint of identified variants (T0: baseline, T1: after 3 months, DS: disease status at T1). Columns represent individual plasma samples and are sorted by disease status (DS) and time points. Each gene affected by alterations is reported and sorted by frequency, which indicates the proportion of patients in whom the gene is mutated on the total number of patients (indicated on the left). The different colors represent the different time points (brown for T0, green for T1). The bar plots at the top and right of the Oncoprint indicate the count of events found, respectively, in each sample and in each gene. In the lower part of the Oncoprint, the disease status (DS), hormonal therapy, and CDK4/6i are indicated for each sample. PD: disease progression, SD: stable disease, PR: partial response. (**B**) The box plot shows the Wilcoxon test applied to the MAF% trend of alterations between T0 and T1 in patients who presented PD and patients with SD or PR. The box plot in the lower part of the figure shows the Wilcoxon test applied to the genomic equivalent of T0 and T1 samples in patients with PR or SD/PR. (**C**) Patient response in terms of PD or SD/PR. The cohort was divided into PI3KCA-mutated or WT in the upper KM plot and into lobular, ductal, and NST in the lower KM plot. T0: baseline; T1: collection time after 3 months of CDK4/6i treatment; PD: disease progression; SD: stable disease; PR: partial response; DS: disease status at T1; MAF: molecules allele frequency; WT: wild-type; NST: no special type.

**Table 1 biomedicines-12-02183-t001:** Patient and tumor characteristics.

Variable	(N°)
Patients enrolled	16
Age, median value	72 (38–75)
Histology	
Lobular	8
Ductal	5
NST	3
Hormonal Therapy	
Fulvestrant	5
Letrozole	11
CDK4/6i	
Abemaciclib	3
Palbociclib	5
Ribociclib	8
Response at T1	
PD	5
SD	3
PR	8

NST: no special type; CDK4/6i: CDK4/6 inhibitors; T1: collection time after 3 months of CDK4/6i treatment; PD: disease progression; SD: stable disease; PR: partial response.

## Data Availability

The original contributions presented in the study are included in the article/Appendix A, further inquiries can be directed to the corresponding author.

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
