# Peer review of "A Novel AKT1, ERBB2, ESR1, KRAS, PIK3CA, and TP53 NGS Assay: A Non-Invasive Tool to Monitor Resistance Mechanisms to Hormonal Therapy and CDK4/6 Inhibitors"

_biomedicines, 2024, doi:10.3390/biomedicines12102183_

Round 1

Reviewer 1 Report

Comments and Suggestions for Authors

The manuscript by Alessandra Virga et al describes the feasability of a sequencing approach called Plasma-SeqSensi NGS to monitor genetic alterations in a specific cancer associated gene panel ( AKT1, ERBB2, ESR1, KRAS, PIK3CA, and TP53) in circulating cell free DNA of patients with metastatic breast cancer. They analyzed samples from 16 patients with HR+/HER2- tumors at baseline and after 3 months of treatment with endocrine therapy and CDK4/6 inhibitors. Mutant allele frequency appeared to be associated with clinical outcome. The study indicates very intersting and highly relevant results. However, there are minor points that should be addresed prior publication:

In the Material & Methods section it is described that the software classified the somatic vs germline alterations, but what was the criteria set in the software in order to classify?

Further, the cfDNA concentration obtained from 2 ml plasma would be interesting. What was the input (per sample) for the NGS?  What was the quality check based on? The authors gained a good quality check in 96% of the samples, but what does this mean? In the Suppl File  Q30???

Which reference genome was used to identify variants (PQS technology)?

The authors describe HR+/HER2- metastatic breast cander. Does this classification refer to the tissue obtained at diagnosis of the primary tumor or the metastatic site? I would suggest to add the metastatic site in the table as well as nodal status and Ki-67.

Fig. 1 A shows an onco-print which is difficult to interpret and the picture quality should be improved as it appears blurry. What doeas the percentage refer to? Axis labeling is missing.

Fi. 2B must be improved as it is difficult to read. “Condition” should be eithet clinical outcome or time point. The content is interesting, though. What exactly is genomic equivalent?  The authors should describe the principle of it (as used by the software).

Fig 2C) Does the PIK2CA status solely refer to the cfDNA testimg or was tumor tissue analyzed as well?

Suppl. Figure S2 was not attached. I woul like to look at the survival data of the online resources.

It should be discussed that only 2 of 3 patients with SD had detectable variants and 4 of 8 pts with Partial Response.

The authors should reconsider the title as it refers to a specific service by sysmex. Was the study funded by the company? 

Reviewer 2 Report

Comments and Suggestions for Authors

In Manuscript entitled Plasma-SeqSensei NGS assay: a non-invasive tool to monitor resistance mechanisms to hormonal therapy and CDK4/6 inhibitors written by authors Alessandra Virga * , Caterina Gianni , Michela Palleschi , Davide Angeli , Filippo Merloni , Roberta Maltoni , Paola Ulivi , Giovanni Martinelli , Ugo De Giorgi , Sara Bravaccini were evaluated the accuracy and feasibility of a novel multi-gene target panel NGS assay to detect genomic alterations of AKT1, ERBB2, ESR1, KRAS, PIK3CA, and TP53 on circulating tumor DNA (ctDNA) in women with BC treated with hormonal therapy (HT) plus CDK4/6 inhibitors (CDK4/6i). Additionally, the authors aim to investigate the association between genomic profiling and clinical outcomes.

The topic is very interesting, but we have some suggestions>

1.      The aim of the study and the conclusion are not compatible. In the aim of the study the authors said that they want to evaluate the accuracy and feasibility of a novel multi-gene target panel NGS assay to detect genomic alterations of AKT1, ERBB2, ESR1, KRAS, PIK3CA, and TP53 on circulating tumor DNA (ctDNA) in women with BC, and to investigate the association between genomic profiling and clinical outcomes. In the conclusion the authors state that Liquid biopsy analysis by PSQ is feasible and accurate. According to literature data, the PIK3CA alterations were mainly seen in patients who developed CDK4/6i’s resistance. Increased and decreased MAF levels have a key role in treatment monitoring.”

2.      The number of samples are very small for some serious conclusion, so you need to expand your research and add some addition samples.

Comments on the Quality of English Language

In Manuscript entitled Plasma-SeqSensei NGS assay: a non-invasive tool to monitor resistance mechanisms to hormonal therapy and CDK4/6 inhibitors written by authors Alessandra Virga * , Caterina Gianni , Michela Palleschi , Davide Angeli , Filippo Merloni , Roberta Maltoni , Paola Ulivi , Giovanni Martinelli , Ugo De Giorgi , Sara Bravaccini were evaluated the accuracy and feasibility of a novel multi-gene target panel NGS assay to detect genomic alterations of AKT1, ERBB2, ESR1, KRAS, PIK3CA, and TP53 on circulating tumor DNA (ctDNA) in women with BC treated with hormonal therapy (HT) plus CDK4/6 inhibitors (CDK4/6i). Additionally, the authors aim to investigate the association between genomic profiling and clinical outcomes.

The topic is very interesting, but we have some suggestions>

1.      The aim of the study and the conclusion are not compatible. In the aim of the study the authors said that they want to evaluate the accuracy and feasibility of a novel multi-gene target panel NGS assay to detect genomic alterations of AKT1, ERBB2, ESR1, KRAS, PIK3CA, and TP53 on circulating tumor DNA (ctDNA) in women with BC, and to investigate the association between genomic profiling and clinical outcomes. In the conclusion the authors state that Liquid biopsy analysis by PSQ is feasible and accurate. According to literature data, the PIK3CA alterations were mainly seen in patients who developed CDK4/6i’s resistance. Increased and decreased MAF levels have a key role in treatment monitoring.”

2.      The number of samples are very small for some serious conclusion, so you need to expand your research and add some addition samples.

Reviewer 3 Report

Comments and Suggestions for Authors

The manuscript presents the analysis of  6 genes in circulating tumor DNA in 16 breast cancer patients, 5 with progression disease, 3 presented stable disease and 8 had partial response. The authors applied NGS sequencing to evaluate the presence of mutations in panel of six genes: AKT1, ERBB2, KRAS, PIK3CA and TP53.

Genomic equivalency and MAF significantly differed between T0 and T1 stage in patients with stable or progression disease. They found also that mutations in PIK3CA and ESR1 genes were found mainly in patients with progression disease.

The study is very small so it is not possible to draw conclusions what the authors underline.

Reviewer 4 Report

Comments and Suggestions for Authors

This is an original article that investigate the association between genomic profiling and clinical outcomes in metastatic breast cancer patients.

it is a well presented study, with up to date references, no self-citations, no ethical concerns.

Tables and firgures are fine. Language is appriopate. No plagiarism was detected.

My concern is that no enough effort has been given to the discussion section, which is the most important section of the manuscipt. In discussion all important study data should be mentioned and explained, together with literature review.
